Alu hypermethylation and high oxidative stress in patients with musculoskeletal tumors

Woraruthai Thamonwan 1
Charoenlap Chris 2
Hongsaprabhas Chindanai 2
Mutirangura Apiwat 3
Honsawek Sittisak sittisak.h@chula.ac.th 1
1 Osteoarthritis and Musculoskeleton Research Unit, Department of Biochemistry, Faculty of Medicine, King Chulalongkorn Memorial Hospital, Thai Red Cross Society, Chulalongkorn University , Bangkok , Thailand
2 Department of Orthopaedics, Vinai Parkpian Orthopaedic Research Center, Faculty of Medicine, King Chulalongkorn Memorial Hospital, Thai Red Cross Society, Chulalongkorn University , Bangkok , Thailand
3 Center for Excellence in Molecular Genetics of Cancer & Human Diseases, Department of Anatomy, Faculty of Medicine, King Chulalongkorn Memorial Hospital, Thai Red Cross Society, Chulalongkorn University , Bangkok , Thailand
Albertini Maria Cristina
Electronic publication date: 2018 Aug 16
Publication date: 2018
Volume: 6
Electronic Location ID: e5492
Received 2018 Mar 9; Accepted 2018 Jul 27
Copyright: ©2018 Woraruthai et al.
Copyright year: 2018
Copyright holder: Woraruthai et al.
License: This is an open access article distributed under the terms of the Creative Commons Attribution License, which permits unrestricted use, distribution, reproduction and adaptation in any medium and for any purpose provided that it is properly attributed. For attribution, the original author(s), title, publication source (PeerJ) and either DOI or URL of the article must be cited.
License URL: https://creativecommons.org/licenses/by/4.0/

Keywords: Alu methylation, 8-OHdG, Blood leukocytes, Oxidative stress, Musculoskeletal tumors

Funding: 90th Anniversary of Chulalongkorn University Ratchadaphiseksomphot Endowment Fund of Chulalongkorn University Thailand Research Fund DPG5980005 This work was supported by the 90th Anniversary of Chulalongkorn University and the Ratchadaphiseksomphot Endowment Fund of Chulalongkorn University, and by the Thailand Research Fund (DPG5980005). The funders had no role in study design, data collection and analysis, decision to publish, or preparation of the manuscript.

==============================
Background

Alu is one of the non-autonomous element retrotransposons, constituting nearly 11% of the human DNA. Methylation changes of the Alu element can cause genomic instability, a hallmark of cancer development, ultimately leading to the development of cancer. Epigenetic factors may induce the aberrant methylation of Alu and also oxidative stress. However, current knowledge of Alu methylation and oxidative stress is limited. There are few studies that have evaluated Alu methylation and oxidative stress on musculoskeletal tumor progression. Therefore, the present study evaluated the status of Alu methylation in musculoskeletal (MS) tumor, adjacent tissues, and blood leukocytes from MS tumor subjects, as well as unaffected participants. Moreover, we also investigated the oxidative stress status in MS tumor subjects and the control participants and determined the correlation between Alu methylation in MS tumors and that in blood leukocytes.

Methods

Musculoskeletal tumors from musculoskeletal tumor patients (n = 40) were compared to adjacent tissues (n = 40). The blood leukocytes from musculoskeletal tumor patients were compared to the blood leukocytes from controls (n = 107). Alu methylation status was analyzed using quantitative combined bisulfite restriction analysis (COBRA). In addition, 8–hydroxy 2′–deoxyguanosine (8–OHdG) values were determined using enzyme—linked immunosorbent assay.

Results

Alu methylation values in MS tumors were statistically significantly higher than those in adjacent tissues (P = 0.035). Similarly, Alu methylation statuses in the blood leukocytes of MS tumor subjects were statistically greater than those of control participants (P < 0.001). Moreover, there was a positive association between Alu methylation levels in MS tumors and blood leukocytes (r = 0.765, P < 0.001). In addition, the highest tertile was significantly associated with the risk of MS tumors (OR = 14.17, 95% CI [5.08–39.51]; P < 0.001). The 8-OHdG values in MS tumors were statistically higher than in adjacent tissues (P < 0.001) and circulating 8-OHdG levels were substantially greater in MS tumor subjects than in the control participants (P < 0.001).

Discussion

These findings suggest that Alu methylation in blood leukocytes and plasma 8-OHdG might represent non-invasive biomarkers to help diagnose MS tumors. Therefore, Alu hypermethylation and high oxidative stress might be involved in the pathogenesis of the musculoskeletal tumors.

Introduction

Musculoskeletal (MS) tumors are uncommon and distinct, as compared to other tumors. Osteogenic and chondrogenic sarcomas account for approximately 0.5% of all malignancies in humans. Osteosarcoma affects mainly children and adolescents but most chondrosarcoma occurs predominantly in adults. The incidence of soft tissue sarcomas is three- to fourfold greater, and the majority of these patients are observed mainly after the 5th decade of life (Ofluoglu et al., 2010). The etiologies of MS tumors remain far from clear. Genetic and epigenetic factors could be involved in the pathogenesis of many tumors (Chalitchagorn et al., 2004). Moreover, a recent study suggests that epigenetic changes can be influenced by various factors such as age, environment, lifestyle, and disease states that can be related to tumor progression (Pogribny & Beland, 2009).

There are three main processes of epigenetic change. One of these processes is DNA methylation; the cytosine base is modified at the C5 position (5mC) which inhibits the transcriptional process. However, this mechanism regulates the gene expression in the human genome, inactivates the X chromosome, affects the embryonic development, and most notably, suppresses the retrotransposon activity (Jin, Li & Robertson, 2011). A methylation site in humans establishes retrotransposon elements including short interspersed nuclear elements or Alu, long interspersed nuclear elements-1 (LINE-1), and SAT-1. Alu is a non-autonomous retrotransposon, constituting nearly 11% of human DNA (Deininger et al., 2003). It can promote imperfect chromosome recombination, or insertions or deletions of the chromosomes (Mighell, Markham & Robinson, 1997). Moreover, Alu can be inhibited by any DNA methylation process throughout its nucleotide sequences. A growing body of evidence suggests that methylation changes in Alu can cause genomic instability, a hallmark of cancer development, ultimately leading to the progression of cancer (Bae et al., 2012; Saito et al., 2010). Previous studies have shown that Alu methylation of various cancers is lower than in control groups (Weisenberger et al., 2005; Rhee et al., 2015; Jordà et al., 2017). However, a study reported that the percentage of Alu methylated samples compared to a reference (PMR) in white blood cells from pancreatic cancer patients was higher than in healthy controls (Neale et al., 2014). A previous study revealed that methylation of retrotransposons was involved in the oxidative stress process which may be associated with musculoskeletal tumor progression (Donkena, Young & Tindall, 2010).

Oxidative stress is an imbalance of radicals and antioxidants that generate reactive oxygen species (ROS) in metabolic processes (Barzilai, Rotman & Shiloh, 2002). ROS can cause DNA damage, including mutations, base modification, and DNA strand breakages (Valko et al., 2004). However, ROS cannot be disposed of by the body resulting in a diverse range of diseases, especially cancer (Halliwell & Cross, 1994). One form of DNA damage is the 8–hydroxy 2′–deoxyguanosine (8–OHdG), which is activated by oxidative DNA damage leading to base modification. 8-OHdG adducts can result in G-to-T transversions and mutations and the presence of 8-OHdG adducts in CpG islands strongly suppresses methylation of the adjacent cytosine (Franco et al., 2008). Thus, 8-OHdG can contribute to the aberrant DNA methylation process, leading to alteration of gene expression, genomic instability, and subsequently cancer progression (Ziech et al., 2011). Hence, 8-OHdG has widely served as a biomarker for oxidative stress. Current knowledge suggests that both oxidative stress and DNA methylation are factors that lead to diverse groups of cancer. Moreover, 8-OHdG levels can inhibit the DNA methylation process at the cytosine base, causing DNA hypomethylation. However, in the promoter region, 8-OHdG can act as a catalyst for DNA methylation, leading to DNA hypermethylation (Franco et al., 2008). In hepatocellular carcinoma, higher 8-OHdG levels were related to hypermethylation of tumor suppressor genes (Nishida et al., 2013). Based on these observations, Alu methylation in the human genome may be induced by increased oxidative DNA damage.

There is limited information on the association between Alu methylation and oxidative stress in musculoskeletal tumor progression. Therefore, the present study evaluated the Alu methylation status in MS tumors, adjacent tissues, and blood leukocytes from MS tumor subjects and unaffected volunteers. Moreover, we also investigated the oxidative stress status in MS tumor subjects compared with the control participants and determined the correlation between Alu methylation in musculoskeletal tumors and that in blood leukocytes.

Materials & Methods

Study population

The experimental protocols were affirmed by the Ethical Committee on Human Research (IRB No. 439/59) of our institution. Written inform consent was provided by all participants in this study.

This cross-sectional study had 40 MS tumor patients between the ages 18–80 years who had surgical treatment at the Department of Orthopedics, Chulalongkorn Memorial Hospital. MS tumor specimens were collected from 11 patients with liposarcoma, five patients with leiomyosarcoma, four patients with giant cell tumor, four patients with osteosarcoma, four patients with chondrosarcoma, four patients with spindle cell tumor, four patients with lipoma, and four patients with schwanoma. None of them had received preoperative systemic chemotherapy treatment. Tissues from MS tumors and non-neoplastic adjacent tissues and venous blood samples were collected at the time of surgery. Histologically-normal, non-neoplastic adjacent tissues were at least 2 cm from the tumor margin. Venous blood samples were also collected from 107 healthy controls between the ages 50–65 years who came for their annual health check-up at the hospital. Tissues, including non-neoplastic adjacent tissues and neoplastic tissues, plasma, and blood leukocytes were collected from all participants and stored at −80 °C.

DNA extraction and quantitative combined bisulfite restriction analysis (COBRA)

Genomic DNA was isolated from tissues and peripheral blood mononuclear cells using a GF-1 nucleic acids extraction kit (Vivantis, Buckinghamshire, Malaysia). The concentration of DNA samples was measured using a Nanodrop 2000 spectrophotometer (Thermo Scientific, Wilmington, DE, USA). Then, 50 ng of the extracted DNA was treated with bisulfite treatment using an EZ DNA Methylation- Gold™ Kit (Zymo Research Corporation, Orange, CA, United States), according to the manufacturer’s protocol.

Alu methylation levels and patterns were analyzed using a COBRA Alu protocol (Udomsinprasert et al., 2016). COBRA is a standard assay for determining Alu methylation status (Jintaridth & Mutirangura, 2010).  The Alu sequence primers were based on the nucleotide sequences in the regulatory region of the Alu sequence (the Alu Sx subfamily) (Batzer & Deininger, 2002). Briefly, bisulfited DNA samples were amplified with two specific primers for COBRA Alu (Tiwawech et al., 2014), 5-GGRGRGGTGGTTTARGTTTGTAA-3 and 5-CTAACTTTTTATATTTTTAATAAAAA CRAAATTTCACCA-3. Each sample was quantitated using the polymerase chain reaction (PCR), containing 10X PCR buffer (Qiagen, Hilden, Germany), 200 mM dNTPs (Applied biosystem, United States), 25 mM MgCl2 (Qiagen, Hilden, Germany), 20 µM primers (forward and reverse primers), 0.5 U HotStar Taq DNA polymerase (Qiagen, Hilden, Germany), and 50 ng bisulfited DNA. The PCR program was as follows: initial activation at 95 °C, followed by 40 cycles of 95 °C for 45 s (denaturation process), 57 °C for 45 s (annealing process) and 72 °C for 45 s (extension process), and the last step was at 72 °C for 7 m (final extension process). After the amplification process, the PCR product was digested with 2U Taq I restriction enzyme (Thermo Fisher Scientific, Waltham, MA, USA) in Taq I buffer (Thermo Fisher Scientific, Waltham, MA, USA) and incubated at 65 °C overnight. The cut PCR product was separated on 8% non-denaturing polyacrylamide gels and followed by ethidium bromide staining. The intensities of the DNA fragments were quatified using a Molecular Imager® Gel-Doc with Image Lab™ Software (BioRad, Begonia Straat, Belgium).

Alu methylation analysis

The COBRA Alu was classified into 4 forms according to its methylation value of the two CpG dinucleotides hereinafter: the hypermethylated form (mC mC), the hypomethylated form (uC uC), and two forms of partial methylation (mC uC and  uC mC). Alu methylation levels and patterns were analyzed to examine the actual percentage of methylated CpG dinucleotides. To calculate Alu methylation status, the percentage of Alu methylation levels and patterns were analyzed in each group according to the intensity of the COBRA-digested Alu products. The amplicons obtained after enzymatic digestion of COBRA-Alu products were classified into six bands (133, 90, 75, 58, 43, and 32 bp) which indicated distinct methylation statuses. The percentage of DNA methylation within Alu was computed as the following: A = intensity of the 133-bp band divided by 133; B = intensity of the 58-bp band divided by 58; C = intensity of the 75-bp band divided by 75; D = intensity of the 90-bp band divided by 90; E = intensity of the 43-bp band divided by 43; and, F = intensity of the 32-bp band divided by 32. The Alu methylation patterns were then determined as the following: percentage of overall methylation loci %mC=100×E+B∕2A+E+B+C+D;mCmC=100×F∕A+C+D+F;

%uCmC=100×C∕A+C+D+F;%mCuC=100×D∕A+C+D+F;and

%uCuC=100×A∕A+C+D+F.

8–hydroxy 2′–deoxyguanosine (8–OHdG) enzyme-linked immunosorbent assay (ELISA)

Total DNA was extracted directly from tissues that were homogenized with liquid nitrogen and added to 10 µL proteinase K and 400 µL lysis buffer containing 50 mM tris-hydrochloride (pH 7.4), 1 mM ethylenediaminetetraacetic acid (EDTA) (pH 8.0), 0.5% w/v sodium dodecylsulfate, and incubated at 50 °C for 2 h. After incubation, the lysates of the tissues were added to 250 µL phenol and 250 µL chloroform:indole-3-acetic acid (CHCl3:IAA; 49:1). The lysates were centrifuged at 13,500 rpm for 30 m at 4 °C. The supernatants were carefully transferred into a tube containing 4 µL glycogen, 40 µL sodium acetate, 800 µL absolute ethanol and kept overnight at −20 °C. The lysates were then centrifuged at 13,500 rpm for 30 m at 4 °C. The supernatants were washed with 1 mL 70% ethanol and centrifuged at 13,500 rpm for 5 m at 4 °C. The supernatants were transferred into microcentrifuge tubes and put into a vacuum machine for 15 m before being dissolved with distilled water. The total DNA concentration was determined using a NanoDrop® ND-100 Spectrophotometer (Thermo Fisher Scientific, Waltham, MA, USA) and adjusted to 200 µg/mL DNA in each sample. Fasting venous blood samples from participants were collected in EDTA tubes to facilitate the isolation of plasma and leukocytes, and were then stored at −80 ∘C, if not measured immediately. 8–OHdG values could be assessed using total DNA from tissue lysates and plasma of the participants and healthy controls by a commercially available HT 8-oxo-dG ELISA Kit II (Trevigen, Gaithersburg, MD, USA), based on the instructions of the manufacturer. Antibodies specific to 8-OHdG generated by the entire immunogen were used. Twofold serial dilutions of 8-OHdG standard with a concentration of 0.89–56.7 ng/mL were applied as standards. Subsequently, the absorbance of the samples was measured at 450 nm using microplate reader. A standard optical density-concentration curve was constructed for determination of 8-OHdG concentration in samples. Intra-assay and inter-assay precisions were less than 10% and 15%, respectively. The sensitivity of this assay was 0.57 ng/mL.

Statistical analysis

Alu methylation levels between neoplastic and non-neoplastic adjacent tissues were determined by the Wilcoxon signed-rank test. Alu methylation in blood leukocytes and healthy control subjects was analyzed by the Mann–Whitney U-test. Moreover, the relationship between Alu methylation status in neoplastic tissues and blood leukocytes was analyzed by Spearman’s rank correlation coefficient test. Linear regression analysis was conducted to analyze the potential of Alu methylation levels as a biomarker. In addition, the relationship of Alu methylation levels and musculoskeletal tumor risk was determined using logistic regression adjusted for age, gender, and tumor size. The Mann–Whitney U- test was performed to determine the 8-OHdG values between groups and Kruskal–Wallis H-test for continuous variables. A receiver operating characteristic (ROC) curve was created to examine the area under the curve (AUC) for assessing the practicability of applying Alu methylation level in blood leukocytes as a possible parameter in distinguishing MS tumors subjects from the control participants. Statistical analysis was performed using the Statistical Package for the Social Sciences (SPSS) version 20.0 (SPSS, Inc., Chicago, IL, United States) and figures were constructed using GraphPad Prism version 7.0. Statistical significance was set at P values <0.05.

Results

Hypermethylation of Alu in patients with musculoskeletal tumors

The level of Alu methylation was assessed in 40 MS tumors, 40 adjacent tissues, and 107 healthy controls. Alu methylation levels were examined in MS tumors compared to the adjacent tissues, and Alu methylation in the blood leukocytes of MS tumor patients was compared to the control subjects using COBRA Alu. The results revealed that the Alu element could be divided into six fragments: 133, 90, 75, 58, 43, and 32 bp (Fig. 1).

Figure 1 Alu methylation of participants.

8% non-denaturing polyacrylamide gel of Alu elements in adjacent tissues, MS tumors and blood leukocytes, respectively (M, 25 bp DNA marker, A, adjacent tissues, T, MS tumors and B, blood leukocytes).

When comparing Alu methylation levels in MS tumors with adjacent tissues, the results showed that median Alu methylation status in MS tumors was statistically greater than that in the adjacent tissues (63.95% vs 58.84%, P = 0.035) (Fig. 2A). Similarly, median Alu methylation in blood leukocytes of MS tumor subjects was statistically greater than in the control participants (71.23% vs 55.95%, P < 0.001) (Fig. 2B).

Figure 2 Percentage of Alu methylation levels in MS tumor patients and controls.

(A) Alu methylation levels in adjacent tissues and MS tumors, (B) Alu methylation levels in blood leukocytes of MS tumor patients compared with controls.

Association of Alu methylation levels in MS tumors and blood leukocytes

Alu methylation levels between MS tumors and blood leukocytes were analyzed to determine if it could be used as a non-invasive biomarker. Interestingly, the results revealed that there was a positive correlation between Alu methylation levels in MS tumors and blood leukocytes (r = 0.765, P < 0.001) (Fig. 3). Moreover, a ROC curve was evaluated to distinguish blood leukocytes from patients with MS tumors from the controls. The area under the curve (AUC) for Alu methylation in blood leukocytes was 0.832 (95% CI [0.746–0.918], P < 0.001) (Fig. 4). This model has a threshold cut-off value of 60.274, sensitivity of 0.825 and specificity of 0.759.

Figure 3 Positive correlation between Alu methylation levels in MS tumor and blood leukocytes.

Each data point indicates the correlation between Alu methylation levels in MS tumors and blood leukocytes.

Figure 4 ROC curve analysis of Alu methylation levels in blood leukocytes that distinguish MS tumors from controls.

Relationship of Alu methylation and susceptibility of MS tumors

To evaluate Alu methylation status in peripheral blood mononuclear cells and the susceptibility to MS tumors, logistic regression was performed. As displayed in Table 1, the results indicated that overall Alu methylation levels increased, reflecting a higher occurring risk for MS tumors (OR = 1.13, 95% CI [1.08–1.18], P < 0.001). Furthermore, Alu methylation distribution was categorized into low methylation and high methylation. Individuals with high Alu methylation above the cut-off value (>60.274) were associated with a 12.8-fold (95% CI [5.46–30.02], P < 0.001) increased risk of MS tumors, compared to individuals with low Alu methylation. When a dose–response effect was determined by tertile, the results showed that Alu hypermethylation was associated with an increased risk for MS tumors. In addition, the highest tertile was significantly associated with the risk for MS tumors (OR = 14.17, 95% CI [5.08–39.51], P < 0.001).

Table 1 Association between Alu methylation and risk of MS tumor.

P-value < 0.05 indicates a significant difference of Alu methylation levels between MS tumor patients and controls.

Repetitive elements	MS tumor (N = 40)	Controls (N = 107)	OR (95% CI)	P-value	
Alu element	
Overall	100%	100%	1.13 (1.08–1.18)	<0.001	
By cut-off value	
Low methylation	19.56%	75.70%	1.00 (reference)		
High methylation	80.44%	24.30%	12.80 (5.46–30.02)	<0.001	
By tertile	
1st tertile	27.10%	33.33%	1.00 (reference)		
2nd tertile	25.80%	33.33%	1.17 (0.36–3.74)	0.796	
3rd tertile	41.10%	33.33%	14.17 (5.08–39.51)	<0.001	

Figure 5 8-OHdG levels in MS tumor patients and controls.

(A) 8-OHdG levels in MS tumors and adjacent tissues, (B) plasma 8-OHdG levels in MS tumor patients compared with controls.

Elevated levels of 8 -OHdG in tumor specimens and plasma from MS tumor patients

In this study, plasma 8-OHdG levels were evaluated in 107 controls and 40 MS tumor patients. 8-OHdG levels were assessed in the tissues and plasma. The results revealed that the median 8-OHdG value in MS tumor specimens was substantially elevated compared with the adjacent tissues (P < 0.001) (Fig. 5A). Similarly, the median plasma 8–OHdG level in MS tumor subjects was also markedly greater than in the control participants (P < 0.001) (Fig. 5B). However, there was no correlation between Alu methylation levels and 8-OHdG levels in the tissues and plasma of MS tumor patients.

Discussion

Research in recent years has shown that epigenetic changes lead to pathogenesis in many cancers. In this study, we investigated the Alu methylation levels in MS tumors and adjacent tissues and in blood leukocytes of MS tumor patients and healthy controls. In addition, this study determined the 8-OHdG levels in MS tumors and adjacent tissues as well as plasma 8-OHdG levels. This study showed that the Alu methylation levels in MS tumors were higher than in adjacent tissues. Also, the Alu element was hypermethylated in blood leukocytes of MS tumor patients. Moreover, there was a positive relationship between Alu methylation status in peripheral blood leukocytes and MS tumors. Notably, Alu methylation values in peripheral blood leukocytes were associated with MS tumor risk. As for the 8-OHdG levels, the results showed that 8-OHdG levels in MS tumors and plasma of patients with MS tumors were higher than in the controls. Therefore, these results propose that epigenetic alteration may play a role in MS tumor progression.

A recent study revealed that repetitive elements methylation can cause alteration of gene expression, leading to genomic instability (Udomsinprasert et al., 2016). Currently, our study is the first that has investigated Alu methylation levels in tumors and blood leukocytes of MS tumor patients and also determined a correlation between blood leukocytes and MS tumors. Moreover, both 8-OHdG levels in plasma and MS tumors were determined. Previous investigations have reported that aberrant methylation is associated with several types of cancers, including lung, breast, colon, and gastric cancers (Kim et al., 2014). Hypomethylation of repetitive elements has been shown to be associated with an increased risk of several types of cancers, whereas other investigations revealed that hypermethylation of these elements were associated with cancer. For example, the Alu methylation levels in white blood cells of colorectal cancer patients were higher than those in the controls (Walters et al., 2013). Likewise, the Alu methylation status in white blood cells of pancreatic cancer patients was also higher than in the controls (Neale et al., 2014). These findings demonstrated that the elevated percentage of methylated samples compared to a reference Alu methylation status in cancer subjects was associated with pancreatic cancer risk (Neale et al., 2014). Using other assays to determine the DNA methylation levels has discovered that the global methylation levels in peripheral blood DNA are significantly higher in breast cancer patients than in the controls (Xu et al., 2012). In addition, LINE-1 methylation levels are significantly higher in renal cell carcinoma patients compared to the controls, and LINE-1 hypermethylation is positively related with an elevated risk of renal cell cancer (Liao et al., 2011). Furthermore, endogenous double-strand breaks in cancer cells contain higher DNA methylation levels than the cellular genome (Pornthanakasem et al., 2008). The relationship between Alu methylation in blood leukocytes and MS tumor risk may be associated with increased methylation around the DNA double-strand break region. However, the majority of the previous investigations have reported that DNA hypomethylation was related to the risk of several type of cancers such as ovarian cancer (Akers et al., 2014), head and neck cancer (Langevin et al., 2012), and breast cancer (Wu et al., 2012). These findings report that repetitive elements in patients with cancer are hypomethylated when compared to the controls. Previous studies have also demonstrated that Alu methylation levels in blood leukocytes of breast and pancreatic cancer patients are significantly lower than those of controls (Wu et al., 2012; Dauksa et al., 2012). Additionally, aberrant DNA methylation (either hypermethylation or hypomethylation) can lead to genomic instability and cancer progression (Esteller, 2007). The possible explanation for this discrepancy remains unclear and it may be due to the differences in cancers, cell types, and/or methodologies used to assess the methylation levels. Several methods to measure methylation level include real-time PCR-based methods, sequencing-based methods, and gel electrophoresis-based quantitative methods (COBRA PCR). Our protocol of COBRA PCR targets shorter amplicon sizes of the Alu sequences, which greatly improve yield when amplifying genetic material derived from cells or tissues. We have validated this method by comparing it to pyrosequencing and found that interspersed repetitive sequence-COBRA was very accurate and reliable (Jintaridth & Mutirangura, 2010).

Interestingly, the findings from this study demonstrated that Alu methylation levels of MS tumors were positively correlated with blood leukocytes. We also found that the Alu methylation could be a promising biomarker because it yielded the proper differentiation power between the MS tumors and the controls. By setting the cut-off value and applying it, we could develop an indicator helpful for distinguishing the MS tumor patients from the controls. We do not recommend that this test be used for confirmation of MS tumors because the histopathological study should remain the gold standard for a definitive diagnosis. However, Alu methylation in blood leukocytes might be utilized as a non-invasive blood-based biomarker for monitoring the severity and progression of MS tumors.

DNA hypomethylation is believed to result in chromosomal instability by allowing silenced areas of the genome, such as retrotransposons, to become active (Chen et al., 1998). Conversely, our findings revealed that Alu hypermethylation in blood leukocytes was associated with MS tumor risk. Whether it is Alu hypomethylation or hypermethylation that was associated with cancer risk, the mechanism underlying this relationship between aberrant DNA methylation and carcinogenesis remain uncertain. The association we have observed between Alu hypermethylation and MS tumor risk might be related to a high frequency of double-strand breaks/ oxidative DNA damage in individuals with tumor, however, future investigations will be needed to validate this assumption.

We observed, 8-OHdG levels of the plasma from MS tumor patients were higher than in controls. In addition, 8-OHdG levels of tumor tissues were significantly higher than the adjacent tissues. This finding is consistent with a previous study that showed elevated oxidative stress was associated with the mortality of patients and that increased serum insulin-like growth factor-1 levels can prevent musculoskeletal tumors from occurring (Elis et al., 2011). Recent studies have also reported that 8-OHdG values in cancer subjects are greater than in control participants (Mohamadkhani et al., 2017; Ma-On et al., 2017). These results suggest that increased 8-OHdG levels might be a parameter of high oxidative stress, defective antioxidant protection and/or deficient repair of oxidative DNA damage. Furthermore, 8-OHdG levels could suppress DNA methylation at nearby cytosine bases leading to DNA hypomethylation (Wu & Ni, 2015). Moreover, 8-OHdG is potentially one of the most abundant DNA lesions formed by oxidative stress and this mutagenic lesion results in base transversions (Lunec et al., 2002). Therefore, 8-OHdG is involved in the progression of cancer via 8-OHdG adduction. In the current study, there were no correlations between Alu methylation levels and 8-OHdG levels in both plasma and tumor tissues. These findings suggest that oxidative stress was not directly related to repetitive element methylation but oxidative stress could somehow be associated with the pathogenesis of MS tumor.

This is a cross-sectional study, and therefore it has some limitations. First, the MS tumor types were heterogenous. Hence, it is challenging to analyze the Alu methylation levels data for each subgroup. Second, the sample size in this study was relatively small, so the authors cannot conclude with certainty that the Alu methylation levels in each subgroup are higher than in the control group. Additional studies with a larger sample size are required. Third, DNA methylation is an epigenetic mechanism so it is possible that confounding factors such as lifestyle, diet, alcohol drinking, smoking, and body mass index (BMI) may have affected the Alu methylation levels of MS tumor patients. Fourth, the authors could not analyze the association between Alu methylation levels and the clinical outcomes because complete clinical data was not available in our database. Due to the design of this study, there was no long-term follow-up of the MS tumor patients’ symptoms. It is recommended that a prospective study with a long follow-up period could be conducted to further investigate Alu methylation levels and the risk of developing MS tumors. Additionally, incomplete assessment of tumor subtype, tumor stage or grade needs to be taken into consideration due to limitations of record accessibility. Other caveats would be the lack of serum C-reactive protein value. Future study should collect these data to further examine the differences between subgroups.

Conclusion

To summarize, our study illustrated that Alu hypermethylation and elevated 8-OHdG status were evident in MS tumor tissues and that Alu hypermethylation and high oxidative stress were present even in peripheral blood of MS tumor patients. To our knowledge, this study is the first to demonstrate the association between Alu methylation in the MS tumors and in the blood leukocytes, indicating that Alu methylation could be considered as a possible non-invasive blood-based marker when diagnosing MS tumors. We also suggest that Alu hypermethylation might reflect the severity of an epigenetic field for tumorigenesis and could become an epigenetic biomarker for the tumor risk prediction, monitoring, and follow-up of MS tumor patients. Additional research is warranted using prospective cohort designs to affirm this finding, further unravel a causal and/or correlative relationship and to yield more evidence for the utility of examining Alu methylation in blood leukocytes as a potential biomarker of MS tumors.

Supplemental Information

Supplemental Information 1 Raw data of musculoskeletal tumor and controls

Values of Alu methylation and 8-OHdG levels in musculoskeletal tumor, adjacent tissue, blood leukocytes, and controls.

Click here for additional data file.

The authors would like to thank the Department of Biochemistry and Chulalongkorn Medical Research Center (Chula MRC), Faculty of Medicine, Chulalongkorn University, for providing the facilities. The authors acknowledge the staff of Center for Excellence in Molecular Genetics of Cancer & Human Diseases and Nipaporn Theerawattanapong for their technical assistance. We thank June Ohata, Tom Mabey, and Ellie McConachie for reviewing and proof-reading the manuscript.

Additional Information and Declarations

Competing Interests

Author Contributions

Human Ethics

Data Availability

The authors declare there are no competing interests.

Thamonwan Woraruthai conceived and designed the experiments, performed the experiments, analyzed the data, contributed reagents/materials/analysis tools, prepared figures and/or tables, authored or reviewed drafts of the paper, approved the final draft, conceived and designed the experiments, performed the experiments, analyzed the data, wrote the paper, prepared figures and/or tables, reviewed drafts of the paper.

Chris Charoenlap and Chindanai Hongsaprabhas conceived and designed the experiments, performed the experiments, analyzed the data, contributed reagents/materials/analysis tools, authored or reviewed drafts of the paper, approved the final draft, contributed reagents/materials/analysis tools, analyzed the data, reviewed drafts of the paper.

Apiwat Mutirangura conceived and designed the experiments, performed the experiments, contributed reagents/materials/analysis tools, authored or reviewed drafts of the paper, approved the final draft, analyzed the data, contributed reagents/materials/analysis tools, reviewed drafts of the paper.

Sittisak Honsawek conceived and designed the experiments, performed the experiments, analyzed the data, contributed reagents/materials/analysis tools, prepared figures and/or tables, authored or reviewed drafts of the paper, approved the final draft, conceived and designed the experiments, analyzed the data, wrote the paper, provide funding.

The following information was supplied relating to ethical approvals (i.e., approving body and any reference numbers):

The experimental protocols were affirmed by the Institutional Review Board, Faculty of Medicine, Chulalongkorn University (IRB No. 439/59).

The following information was supplied regarding data availability:

The raw data are provided in Supplemental Information 1.

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
