# Peer review of "Alu hypermethylation and high oxidative stress in patients with musculoskeletal tumors"

_PeerJ, doi:10.7717/peerj.5492_

## Round 0.1 · original submission · Major Revisions

Important issues in the manuscript need to be addressed as specified by the reviewers.

·

Basic reporting

The English is not of a standard that is acceptable for publication. There are some statements that are frankly incorrect e.g. osteogenic and chondrogenis sarcoma affect mainly children. Osteosarcoma affects mainly children but chondrosarcoma affects mainly adults. The statement that patients with MS tumor under the age of 20 all being dead is completely inaccurate.
The data is incomplete- there are about 40 patients assessed but there is no information regarding the type of tumours they had or whether they received any other treatment such as chemotherapy.

Experimental design

The research question is clear i.e. whether Alu methylation correlates with outcomes but the importance of this has been overstated. Other blood tests such as CRP correlate with survival in bone sarcoma and can be carried out cheaply.
The methodology does not contain enough information to allow repetition. There is no description how the tissue lysates were derived or how the plasma from blood samples was obtained.

Validity of the findings

The main flaw of this paper is the Alu gel which shows some bands that do not vary between the conditions. The authors then extrapolate these bands to suggest that there is a difference. This is misleading- the bands can only be semi-quantified at best, but the authors do not provide any raw data to demonstrate how they have performed their quantification. The data is NOT robust and is essentially meaningless.

Additional comments

This paper does not contribute anything of value. If experiments are done on tissues and blood, it should at least be explained exactly what tumour sub-types were studied. The methodology is poor and the interpretation of the data is overblown.

Reviewer 2 ·

Basic reporting

The structure of the manuscript is clear and logical, with figures that clearly illustrate the main findings. However, there are some issues with phrasing and the language throughout the manuscript. It would benefit from careful proof-reading and editing, perhaps with assistance from a native English speaker if possible.

Experimental design

The research question is well defined and addressed. However, several aspects of the methods need to be clarified to ensure the work was performed to suitably high standards. COBRA is also not a sensitive technique, and the Alu sequences interrogated by the assay need to be confirmed.

Validity of the findings

The data appears to be robust and appropriately analysed, and the conclusions are supported by the findings.

Additional comments

The work is interesting and suggests that Alu methylation should be explored further in MS tumours, both in terms of mechanistic studies and examination of its potential as a diagnostic/prognostic biomarker. However, some issues in the manuscript need to be addressed.

Major comments
1. Which Alu elements in the genome are being interrogated by the COBRA assay? Alu elements can be categorised by their evolutionary origin, with the widely used pyrosequencing assay specifically interrogating Alu Sx sequences in the human genome. It is known that these subfamilies display differential methylation, such as in response to environmental pollutants (Byun et al, 2013, Part Fibre Toxicol 10:28). Furthermore, only AluY sequences are believed to have retained the ability to retrotranspose within the genome. It is therefore important to appreciate which subfamilies may be analysed by this assay.

2. As noted by the authors, most studies have reported hypomethylation of Alu elements in cancer, which may lead to genomic instability. In the Discussion section the authors should address the possible implications of Alu hypermethylation and how it may be implicated in carcinogenesis.

3. Were Alu methylation and 8-OHdG levels in tumours examined in relation to tumour stage or grade?

4. The authors commented on the heterogenous nature of the MS tumours. A description of the tumour types and numbers of patients should be provided in the Methods section (Line 119 onwards). Furthermore, were differences in Alu methylation and 8-OHdG levels examined between the different subtypes?

5. The analysis of the COBRA data is not well explained. The authors should modify the ‘Alu methylation analysis’ subsection of the Methods (Line 147) to provide a better description of how the results were interpreted, to enable the reader to more easily understand the work that was performed.

6. Where was the adjacent tissue taken from, i.e. what distance from the tumour? Was the tissue examined by a histopathologist to confirm the absence of malignant cells? These are important issues to clarify due to the possibility of field defects in the surrounding normal tissue.

7. Was a standard curve used for the ELISA measuring 8-OHdG levels? If so, this should be clarified in the Methods section (Line 164).

8. Was the analysis of 8-OHdG performed in plasma for both the 107 healthy controls and 40 patients (Line 222 and Figure 5B)? If so, the text should be modified to avoid referring to “tumors”, as this implies analysis of tumour tissue rather than plasma.

9. The AUC from the ROC analysis is reasonably promising and should form part of the discussion of the potential of Alu methylation as a blood-based biomarker of MS tumours (Lines 278-281).

10. The Conclusions paragraph (Line 310) should be re-written so that it does not simply repeat the findings once more.

11. In Figure 3, two outliers seem to have erroneously high methylation levels in leukocytes. Are the authors confident that the results for these two samples are accurate?

12. In the analysis of relative disease risk, why was the methylation data dichotomised by the median value (Line 214), rather than the threshold identified by ROC analysis (Line 208)? Furthermore, the sentence beginning “A high methylation” on Line 215 should be re-written to clarify the meaning.


Minor comments
Line 66: The sentence ending “usually dead” needs to be revised to clarify the meaning and to be more delicately phrased.
Line 95: The sentence beginning “Moreover, 8-OHdG can interfere” should be rewritten to something more specific, such as by describing how 8-OHdG could lead to base changes at CpG sites that eliminate the possibly of methylation.
Line 154: Here the authors state that the assay produces 6 bands, but later they describe the assay as producing 5. This should be corrected.
Line 166: The ELISA kit should be named.
Line 196: The mean/median methylation levels in tumours, adjacent tissue and blood samples should be written in the text, to help the reader understand the magnitude of the changes in methylation being reported.
Line 231: The opening paragraph of the Discussion uses the phrase “to the controls” or “than in the controls” a total of five times, which is too repetitive. These could be removed without affecting how the reader will interpret the paragraph.
Line 257: A reference for the pancreatic cancer study appears to be missing.
Line 258: “PMR” needs to be defined. What is it?
Line 264: The suggestion that the tumours have double stranded breaks that are easily methylated is speculative. Please rephrase this sentence to be more cautious.
Line 273: The sentence beginning “These findings” should be revised to be more cautious, as those results alone do not suggest that Alu hypomethylation leads to genomic instability.
Line 288: Again, more caution must be taken into the inferences made from results. There is nothing to suggest that 8-OHdG could remodel chromatin structure simply from those results.

---

## Round 0.2 · accepted · Accept

I have checked all the responses of the authors and they address the issues well. The manuscript is now ready for publication.